# Impact of statin withdrawal on perceived and objective muscle function

Paul Peyrel[1,2], Pascale Mauriège[1,2], Jérôme Frenette[3,4], Nathalie Laflamme[3], Karine Greffard[3], Sébastien S. Dufresne[5], Claire Huth[1,2], Jean Bergeron[3,6], Denis R. Joanisse[1,2]*

1 Department of Kinesiology, Université Laval, Québec, Québec, Canada, 2 Research Center of the University Institute of Cardiology and Pulmonology of Quebec, Québec, Québec, Canada, 3 CHU de Québec – Université Laval Research Center, Québec, Québec, Canada, 4 Department of Rehabilitation, Université Laval, Québec, Québec, Canada, 5 Department of Health Sciences, Université du Québec à Chicoutimi, Saguenay, Québec, Canada, 6 Department of Laboratory Medicine and of Medicine, Université Laval, Québec, Québec, Canada

* denis.joanisse@kin.ulaval.ca

**Data Availability Statement:** All relevant data are within the paper and its Supporting information files.

## Abstract

### Background and aims

Statin-associated muscle symptoms (SAMS) are frequently reported. Nevertheless, few data on objective measures of muscle function are available. Recent data suggesting an important nocebo effect with statin use could confound such effects. The objective was to assess if subjective and objective measures of muscle function improve after drug withdrawal in SAMS reporters.

### Methods

Patients (59 men, 33 women, 50.3±9.6 yrs.) in primary cardiovascular prevention composed three cohorts: statin users with (SAMS, n = 61) or without symptoms (No SAMS, n = 15), and controls (n = 16) (registered at clinicaltrials.gov, NCT01493648). Force (F), endurance (E) and power (P) of the leg extensors ($_{EXT}$) and flexors ($_{FLE}$) and handgrip strength ($F_{HG}$) were measured using isokinetic and handheld dynamometers, respectively. A 10-point visual analogue scale (VAS) was used to self-assess SAMS intensity. Measures were taken before and after two months of withdrawal.

### Results

Following withdrawal, repeated-measures analyses show improvements for the entire cohort in $E_{EXT}$, $E_{FLE}$, $F_{FLE}$, $P_{EXT}$ and $P_{FLE}$ (range +7.2 to +13.3%, all p≤0.02). Post-hoc analyses show these changes to occur notably in SAMS (+8.8 to +16.6%), concurrent with a decrease in subjective perception of effects in SAMS (VAS, from 5.09 to 1.85). $F_{HG}$ was also improved in SAMS (+4.0 to +6.2%) when compared to No SAMS (-1.7 to -4.2%) (all p = 0.02).

**Funding:** The study was supported by the Canadian Institutes of Health Research (CIHR grant MOP 114917 to DRJ, JB, and JF [https://cihr-irsc.gc.ca]). The funders had no role in study design, data collection and analysis, decision to publish, or preparation of the manuscript.

**Competing interests:** The authors have declared that no competing interests exist.

## Conclusions

Whether suffering from "true" SAMS or nocebo, those who reported SAMS had modest but relevant improvements in muscle function concurrent with a decrease in subjective symptoms intensity after drug withdrawal. Greater attention by clinicians to muscle function in frail statin users appears warranted.

## Trial registration

This study is registered in clinicaltrials.gov (NCT01493648).

## 1. Introduction

Among the pharmacological approaches used to treat hypercholesterolemia, statins are considered the reference lipid-lowering drugs [1]. By significantly reducing cholesterol synthesis through HMG-CoA reductase inhibition, they are commonly used as part of primary or secondary prevention to limit the risk of cardiovascular disease (CVD). Statins are now used by over 200 million people around the world [2], but controversies remain on the nature and prevalence of their side effects. These effects can include the development of diabetes, elevated circulating liver enzymes and joint pain [3, 4], but statin-associated muscle symptoms (SAMS) are the most frequently reported side effects of statin use. Mild myalgia has often been reported by 5–10% of statin users [4, 5], this rate reaching around 20% in a few reports [6, 7].

That statin use could lead to a variety of mild or moderate muscle symptoms is not surprising. Indeed, myopathies and rhabdomyolysis have been documented from statin use [8–10], giving credence to the idea that less severe symptoms are possible or even likely. Nevertheless, recent work has called into question the true prevalence of SAMS, as distinguishing between the harmful effects truly associated with statins and the so-called nocebo effect is not trivial [11–14]. It is thus essential to develop strategies in order to identify the "true" statin-intolerant patients [15]. Among those proposed is the recent "SAMS—Clinical Index" (SAMS-CI) [16] which purports to classify the origin of muscle pain according to whether it is "unlikely", "possibly" or "probably" related to statin use. This tool remains to be validated for use in a broad patient population.

While statins promote an increase in reported muscle complaints, in different muscle groups (especially pectorals, quadriceps, biceps, and deltoids) [17], they do not appear to lead to a systematic decrease in strength or endurance, physical activity level or performance [18–23]. A recent study by Kawai et al. (2018) [24] assessed the physical performances of 1,022 adults aged between 65 and 88 yrs., depending on whether the participants were using statins or not. In this study, significantly lower handgrip strength (respectively 26.1±7.4kg and 28.1±8.5kg, mean ± standard deviation [SD]) and normal walking speed (respectively 1.30±0.24m/s and 1.36±0.26m/s) were observed between statin users compared with non-users. However, when these data were adjusted for SAMS risk factors (age, sex, body mass index [BMI], and number of medications), differences were lost, suggesting that SAMS were contributory to these effects.

The current state of the scientific literature is both limited and contradictory. Some studies suggest that SAMS have no impact on muscle performance [25, 26]. However, as illustrated by Parker et al. (2013), a reduction in performance has been previously demonstrated in patients reporting SAMS in some observational studies [22]. Nevertheless, in the work of Parker et al. (2013), while a decrease in 5 of the 14 performance variables studied did occur in statin

patients reporting SAMS (n = 18), 4 performance variables also deteriorated in patients reporting muscle symptoms in the placebo group (n = 10). In general, the probability that studies examining the muscle function effects of statins include both true SAMS-sufferers and nocebo reporters certainly is a limiting factor in the current understanding of this phenomenon.

The present study therefore aimed first to focus on the effects of statin withdrawal on perceived SAMS and objective muscle performance, and second to explore the impact of this manipulation according to the SAMS-CI category.

## 2. Methods

### 2.1. Participants

Caucasian men (n = 59) and women (n = 33) aged 30 to 60 years (50.3±9.6 yrs., mean ± standard deviation [SD]) and affiliated to the cardiovascular (CV) lipid prevention clinic at the CHU de Québec-Université Laval (CHUL) were enrolled. These were patients treated with statins in primary CV prevention and with normal blood creatine kinase (CK) levels. Two statin groups were formed: a first self-reporting SAMS (SAMS, n = 61) and a second without SAMS (No SAMS, n = 15). A third group of participants not taking statins served as controls (Controls, n = 16). In this work, data are pooled from a pilot study and a main study (Fig 1). In both studies, the inclusion and exclusion criteria and procedures, with a few exceptions mentioned below, were the same. Participants had to be in good general health, sedentary or moderately active (i.e., less than one hour of leisure-time physical activity performed per week) and to present no contraindications for physical function testing. Patients were required

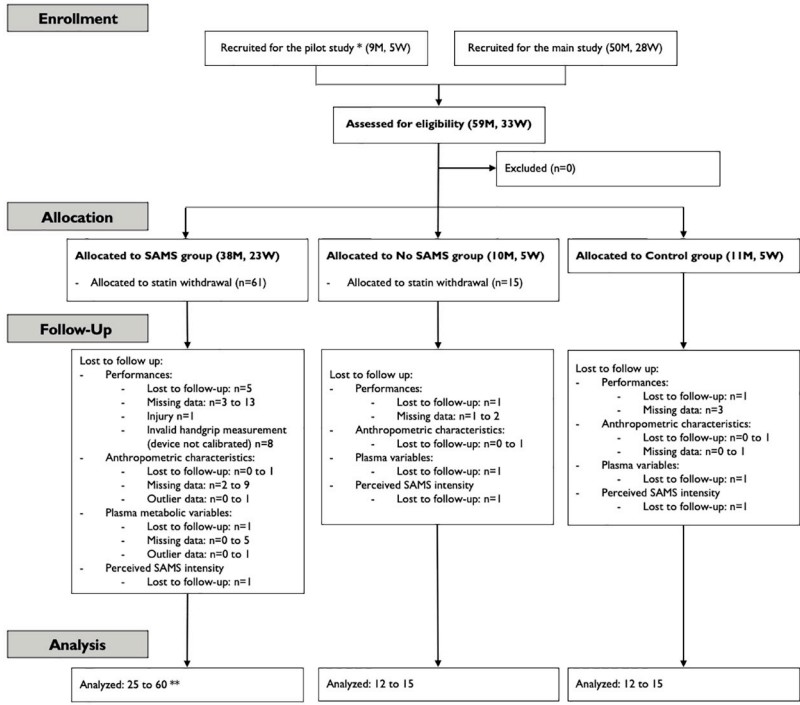

**Fig 1. CONSORT diagram of patients' recruitment.** * All participants recruited into the pilot study had SAMS and were allocated to the SAMS group; ** The large range here is explained by the fact that the participants in the pilot study did not complete all the measures of the study. Exceptions are presented in the methods section; M: men; SAMS: statin-associated muscle symptoms; W: women.

to have a low or moderate Framingham risk score, which allowed the research team to modify their lipid-lowering regimen for the study period. Finally, patients who were previously prescribed other statins or doses prior to their current regimen were not excluded. However, self-reported SAMS needed to be temporally associated with statin use and be present with the current prescription.

Participants were excluded from the study if they had taken other lipid-lowering drugs or any natural treatment that affects lipid metabolism in the last year. Other exclusion criteria included vitamin D deficiency (calcifediol ($25(OH)D_3$) levels below 12.5nmol/L) or vitamin D supplementation; elevated circulating CK levels (>170u/L for women; >195u/L for men) or a history of elevated CK of known or unknown etiology; hepatic or renal failure; untreated hypo- or hyperthyroid; any treatment promoting an increased risk of myopathy; any infection requiring the use of an antibiotic; a daily consumption of more than 60mL of grapefruit juice; hereditary muscle disorders or myopathy, polymyositis or inflammatory myopathy; use of corticosteroids; comorbidities leading to muscle or bone pain (fibromyalgia, arthritis, sensory or intrinsic neuropathy, spinal disease, loss of reflexes, atrophy muscle group); unexplained cramps; cancer in the five years prior to entry into the study; diabetes, stroke, or any known sickle cell trait. In addition, pregnancy, breastfeeding, a physical disability or previous injury interfering with stress testing, depression (within the past three years) or treatment with antidepressants, the use of antipsychotic drugs and alcohol abuse were also exclusion factors.

In a subset of participants (n = 22) for which data were available, comparisons based on SAMS-CI category were performed for SAMS-reporting participants [16].

The protocol was approved by the CHU de Québec–Université Laval ethics committee, and all participants provided informed written consent. The study design complied with the principles of the Declaration of Helsinki and was registered in clinicaltrials.gov (NCT01493648). This study was conducted at the CHU de Québec–Université Laval (CHUL), with enrollment and data acquisition from October 2011 to October 2015.

## 2.2. Procedures

Participants first participated in an inclusion visit. Height (stadiometer), body mass (calibrated scale), BMI ($kg/m^2$), and waist circumference (WC) were assessed (weight, BMI, and WC were not measured after statin withdrawal in the pilot study). Participants completed a standardized recruitment questionnaire, based on current statin type and dosage, statins use history, predisposing conditions or factors that could contribute to muscle problems, as well as an assessment of eating habits.

The experimental protocol consisted of two months of statin withdrawal with pre- and post-withdrawal assessments of subjective and objective measures of muscle function and blood levels of several markers of interest. Data were collected by a blinded experimenter. On the day of pre-test, participants were instructed to take their usual dose of statin at their habitual time. The two-month interval was based on previous studies that showed most SAMS appeared or were resolved within this period of the introduction or withdrawal of the medication [5, 27].

## 2.3. Measurements

**2.3.1. Self-reported muscle symptoms.** Participants self-reported the presence of SAMS by describing what symptom(s) they believed to be the result of medication use. These symptoms typically included myalgia, stiffness, weakness, fatigue and/or cramps. They were then asked to rate the intensity of the symptom(s) they reported by answering the following question (translated from French) on a visual analog scale (VAS):

*On a scale of 0 to 10 (10 being the most intense or unbearable), rate the current intensity of the symptom(s) which you believe to be related to statin use.*

**2.3.2. Blood tests.** A venipuncture was performed following a 12-h overnight fast and prior to muscle function tests. The levels of a number of circulating factors were assessed from these samples (Table 1 and S1 Table), including markers of muscle (CK and myoglobin [MB]) and liver (alanine aminotransferase [ALT] and aspartate aminotransferase [AST]) damage, the plasma lipid-lipoprotein profile (total cholesterol [TC]; triglycerides [TG]; high density lipoprotein [HDL]; low-density lipoprotein [LDL]; TC/HDL; apolipoprotein B-100 [APOB-100] and apolipoprotein A1 [APOA1]), and 25(OH)$D_3$. Only ALT, AST and CK were assessed in the pilot study. All were measured using standardized clinical assays at the CHU de Québec–Université Laval clinical laboratory.

**Table 1. Pre- and post-statin withdrawal anthropometric characteristics and metabolic variables.**

| | SAMS | | | No SAMS | | | Controls | | | Time p | ANOVA Category p | T*C p |
|---|---|---|---|---|---|---|---|---|---|---|---|---|
| | Pre | Post | ES | Pre | Post | ES | Pre | Post | ES | | | |
| Men/Women, n | 38/23 | | | 10/5 | | | 11/5 | | | - | 0.87$ | - |
| Age (years) | 53.0 ± 8.4[a] | - | - | 47.6 ± 9.6[ab] | - | - | 42.4 ± 9.4[b] | - | - | - | **<0.01** | - |
| *Anthropometric characteristics* | | | | | | | | | | | | |
| n | 37 to 59 | | | 14 to 15 | | | 14 to 15 | | | | | |
| Height (cm) | 166 ± 13[a] | - | - | 172 ± 8[a] | - | - | 169 ± 8[a] | - | - | - | 0.20 | - |
| Weight (kg) | 80.9 ± 14.7 | 80.8 ± 14.7 | 0.00 | 74.0 ± 12.3 | 73.9 ± 12.0 | -0.01 | 74.8 ± 13.3 | 74.8 ± 13.9 | -0.01 | 0.76 | 0.10 | 0.97 |
| BMI (kg/m$^2$) | 29.2 ± 4.5 | 29.0 ± 4.5 | -0.06 | 25.1 ± 3.2 | 25.1 ± 3.2 | 0.00 | 26.9 ± 5.9 | 26.9 ± 5.9 | 0.00 | 0.56 | **0.01** | 0.66 |
| WC (cm) | 95.4 ± 17.2 | 93.9 ± 10.1 | -0.07 | 86.2 ± 11.1 | 86.4 ± 11.5 | 0.01 | 84.3 ± 13.0 | 85.1 ± 13.8 | 0.05 | 0.93 | **<0.01** | 0.77 |
| *Plasma metabolic variables* | | | | | | | | | | | | |
| n | 41 to 60 | | | 14 | | | 15 | | | | | |
| MB (μg/L) | 29.9 ± 11.0 | 28.5 ± 9.3 | -0.13 | 29.5 ± 10.7 | 27.9 ± 11.9 | -0.18 | 33.7 ± 18.1 | 30.8 ± 10.4 | -0.08 | 0.13 | 0.46 | 0.84 |
| CK (U/L) | 130 ± 76 | 126 ± 90 | -0.11 | 132 ± 62 | 118 ± 46 | -0.17 | 163 ± 112 | 162 ± 88 | 0.06 | 0.19 | 0.25 | 0.65 |
| ALT (U/L) | 30.0 ± 17.6 | 26.9 ± 15.6* | -0.22 | 23.6 ± 12.0 | 19.1 ± 8.2 | -0.30 | 18.6 ± 5.5 | 18.4 ± 6.3 | -0.04 | **<0.01** | **0.02** | 0.20 |
| AST (U/L) | 22.5 ± 5.9 | 21.7 ± 6.4 | -0.17 | 22.0 ± 6.3 | 20.6 ± 5.3 | -0.20 | 19.7 ± 5.1 | 18.5 ± 4.2 | -0.15 | **0.04** | 0.15 | 0.92 |
| TC (mmol/L) | 4.81 ± 0.98 | 6.78 ± 1.34* | 1.73 | 4.60 ± 0.84 | 6.85 ± 1.37* | 1.92 | 5.06 ± 0.83 | 5.21 ± 0.93 | 0.16 | **<0.01** | **0.04** | **<0.01** |
| TG (mmol/L) | 1.89 ± 1.22 | 2.19 ± 1.20 | 0.31 | 1.08 ± 0.54 | 1.66 ± 0.90 | 0.70 | 1.15 ± 1.05 | 1.20 ± 1.07 | 0.03 | **<0.01** | **<0.01** | 0.22 |
| HDL (mmol/L) | 1.31 ± 0.37 | 1.31 ± 0.35 | -0.01 | 1.55 ± 0.42 | 1.49 ± 0.44 | -0.16 | 1.56 ± 0.41 | 1.63 ± 0.42 | 0.16 | 0.93 | **0.03** | 0.08 |
| LDL (mmol/L) | 2.71 ± 0.97 | 4.59 ± 1.31* | 1.86 | 2.56 ± 0.71 | 4.61 ± 1,26* | 2.02 | 3.11 ± 0.92 | 3.18 ± 1.03 | 0.04 | **<0.01** | 0.26 | **<0.01** |
| TC/HDL (mmol/L) | 3.95 ± 1.36 | 5.54 ± 1.80* | 1.10 | 3.09 ± 0.64 | 4.82 ± 1.07* | 1.85 | 3.48 ± 1.26 | 3.45 ± 1.33 | -0.04 | **<0.01** | **<0.01** | **<0.01** |
| APOB-100 (g/L) | 0.93 ± 0.26 | 1.41 ± 0.33* | 1.69 | 0.87 ± 0.17 | 1.36 ± 0.28* | 2.13 | 0.91 ± 0.22 | 0.93 ± 0.28 | 0.05 | **<0.01** | **<0.01** | **<0.01** |
| APOA1 (g/L) | 1.47 ± 0.22 | 1.44 ± 0.24 | -0.11 | 1.56 ± 0.24 | 1.54 ± 0.25 | -0.09 | 1.61 ± 0.23 | 1.60 ± 0.20 | 0.00 | 0.36 | 0.09 | 0.81 |
| 25(OH)$D_3$ (mmol/L) | 74.2 ± 32.5 | 68.4 ± 32.0† | -0.18 | 71.2 ± 27.0 | 63.8 ± 25.0 | -0.25 | 66.9 ± 18.7 | 68.3 ± 19.2 | 0.06 | **0.02** | 0.91 | 0.12 |
| *Self-reported muscle symptoms intensity* | | | | | | | | | | | | |
| n | 60 | | | 14 | | | 15 | | | | | |
| VAS (0 to 10) | 5.09 ± 1.81 | 1.85 ± 2.25* | -1.22 | 0.00 ± 0.00 | 0.00 ± 0.00 | / | 0.00 ± 0.00 | 0.00 ± 0.00 | / | **<0.01** | **<0.01** | **<0.01** |

Data are expressed as mean ± standard deviation;

$ P-value is shown here by a chi-squared test;

Post value with * is statistically different from the pre value with p≤0.01; Post value with † is statistically different from the pre value with p<0.05. ALT: alanine aminotransferase; APOA1: apolipoprotein A1; APOB-100: apolipoprotein B-100; AST: aspartate aminotransferase; BMI: body mass index; WC: waist circumference; CK: creatin kinase; ES: effects size; HDL: high-density lipoprotein; LDL: low-density lipoprotein; MB: myoglobin; SAMS: statin-associated muscle symptoms; TC: total cholesterol; TC/HDL: cholesterol total / HDL cholesterol; TG: triglycerides; VAS: visual analogic scale; 25(OH)$D_3$: Calcifediol.

**2.3.3. Muscle performance.** Muscle performance was assessed using a Biodex isokinetic dynamometer (Biodex Medical Systems, Shirley, NY). The system has shown good reproducibility in the healthy population [28]. Patients were seated in an upright position and the resistance pads were aligned according to the manufacturer's instructions. The maximum force (F), power (P) and endurance (E) of the extensor (EXT) and flexor (FLE) muscles of the dominant leg were measured.

Testing began with a warm-up consisting of five repetitions at 60˚/s including one movement at maximum voluntary contraction (MVC). Subsequently, the maximum force (Nm) was measured during three MVCs each at 60˚/s and 180˚/s. The highest measured value was retained. Endurance and power were measured over 15 MVC repetitions at 180˚/s. Endurance was calculated as the total (sum) of strength developed during all 15 repetitions (Nm). Power was calculated (W) by multiplying the endurance value with the total time required for the patient to complete the repetitions; then divided by the number of repetitions (n = 15). For all tests, each maximum contraction was performed within the first 90 degrees of the knee's range of motion.

**2.3.4. Handgrip strength.** Handgrip strength was measured with a Jamar hydraulic hand dynamometer (Asimov Engineering, Los Angeles, CA). This tool is reliable and validated [29]. Patients performed the test seated with their feet slightly apart, the unassessed arm at their side, the arm assessed at 90˚ and not resting on an armrest. The handle of the dynamometer was adjusted so that the handle rested on the middle of the four fingers while positioning the base of the dynamometer on the first metacarpal. Once in position, the patient was asked to squeeze the handle of the dynamometer as hard as possible for 3s while exhaling. One min of rest between each contraction was respected. The best of the three repetitions was retained to assess handgrip F (kg) in the right ($F_{HGR}$) and left ($F_{HGL}$) hands. In the pilot study, handgrip strength was only measured in the dominant hand.

## 2.4. Statistical analyses

Data are expressed as mean ± SD. Sample size calculations were performed using predicted changes to measures of physical performance following statin withdrawal. Using a one-tailed model with p = 0.05, alpha = 0.8 (beta = 0.2) and aiming for a 20% mean increase in several muscle function tests using observed SD from preliminary data from 9 participants, required total n values were found to range from 15 to 30. Using $F_{EXT}$ as an example of a major outcome and based on preliminary data of 105.8±42.4 Nm (mean ± SD), a total of 27 subjects was calculated to be necessary to statistically detect a 20% increase in this parameter (to 127.0 Nm).

All statistical analyses were performed using JMP software (SAS Institute, Cary, NC). In cases of missing data for any given measure, participants were removed from that analysis. Between group differences for variables not measured repeatedly over time were analyzed by ANOVA followed by Tukey post-hoc tests. A repeated measures factorial design was used to assess differences between groups and over time using the "Full factorial mixed design" add-in for JMP [30]. In this model, fixed effects were group, time, and group*time, and random effects were subject[group] and subject*time[group]. Post-hoc analyses were performed using Tukey tests. Chi-square tests were used to assess distribution differences across groups from contingency tables. These latter tests were not performed when expected observations in a cell fell below n = 5. P-values less than 0.05 were considered statistically significant. Finally, Cohen's d values for pairwise comparisons were used to assess effect sizes (ES) for within-group changes over time and to qualify changes as trivial (Cohen's d <0.2), small (0.2–0.5), moderate (0.5–0.8), or large (>0.8).

## 3. Results

### 3.1. Anthropometric and metabolic variables at baseline

Participants' characteristics are shown in Table 1. Whereas the SAMS group was significantly older (mean +10.6 yrs) than the Control group, the No SAMS group was intermediate (5.4 yrs. younger than SAMS) but not significantly different from the other two groups. Overall, the SAMS group presented a slightly more deteriorated health profile, as revealed by higher BMI (+4.1kg/m$^2$) compared to the No SAMS group. Increased adiposity in the SAMS group was also indicated by a greater WC (mean +11.2 cm, p = 0.01) than that of the No SAMS and Control groups. Despite a significantly higher ALT level in the SAMS group compared to other groups, all markers of muscle and liver injuries (CK, MB, AST, and ALT) were below values of clinical concern. Finally, 25(OH)D$_3$ values did not differ between groups and were not indicative of vitamin D deficiency.

### 3.2. Clinical aspects of statin users

Most statin users in this study were prescribed rosuvastatin (50% of participants) or atorvastatin (35.5%). This distribution was expected given the usual clinical practice at the CHU de Québec–Université Laval lipid clinic. A detailed breakdown of the clinical profile of participants in this study is presented in S2 Table.

The Framingham Score revealed that the CV risk over ten years was low (<10%) for all patients (except for 1 patient [13%, moderate risk]). Nevertheless, the CV risk for the SAMS group was higher than that of the No SAMS and Control groups (mean +1.43% and +1.96%, respectively.) Most participants in the SAMS and No SAMS groups had been using statins for 12 to 48 months (80.4% and 100% respectively). Nineteen participants indicated having previously not tolerated some forms of statins, their distribution being roughly equal in SAMS and No SAMS groups (respectively 31.9% and 26.7% of participants). Among the 38 participants with a family history (first degree) of CVD, 35 were present in groups treated with statins. 91.3% of participants with a family history (first degree) in terms of lipid-lowering treatment were present in these same statin-taking groups. Finally, of the 30 participants in the SAMS group who reported a family history (first degree) of lipid-lowering therapy, 6 also reported a self-reported family history of SAMS.

### 3.3. Statin withdrawal effects on anthropometry and metabolic variables

No changes to anthropometric measures were observed following statin withdrawal in either the SAMS or No SAMS groups (Table 1).

In addition, and as expected, a significant deterioration of the lipid-lipoprotein profile was seen in all statin users following drug withdrawal. Indeed, repeated-measures ANOVA analyses revealed several time x category interactions for most lipids (Table 1). In brief, while no change was observed in any lipid levels studied in the control group in the same period, both SAMS and No SAMS statin users experienced a deteriorated lipid-lipoprotein profile, reflected by increased TC, TG, LDL, TC/HDL, and APOB-100 levels following medication withdrawal. HDL and APOA1 levels remained, however, unchanged.

With respect to markers of tissue damage or dysfunction, while ALT and AST levels are somewhat reduced following withdrawal of the drug, these values always remained well below clinical thresholds. The muscle damage specific markers MB and CK did not differ across groups or change following statin withdrawal and remained well below clinical values of clinical concern throughout the study (Table 1).

Finally, despite a small decrease in $25(OH)D_3$ levels in the statin groups, these values remained not different from those of the control group and were not indicative of vitamin D deficiency. The baseline levels and impact of statin withdrawal on several other plasma variables are presented in S1 Table.

### 3.4. Perceived muscle effects and objective physical performance

As depicted in Table 1, a significant improvement in the perception of muscle symptoms following statin withdrawal was observed in the SAMS group, decreasing by 3.24 units on the 10-point scale.

Results related to objective physical performance showed no time x category interaction for any Biodex isokinetic dynamometer measures. However, significant time effects for all these measures: $E_{EXT}$ (+8.17% overall for the entire group [ES: 0.22, small effect]), $E_{FLE}$ (+11.6% [ES: 0.30, small effect]), $F_{FLE}$ (+7.20% [ES: 0.22, small effect]), $P_{EXT}$ (+9.04% [ES: 0.24, small effect]) and $P_{FLE}$ (+13.3% [ES: 0.34, small effect]), except for $F_{EXT}$ (+3.24%, NS, [ES: 0.11, trivial effect]) (Fig 2). Few between group differences were seen at baseline, with only $E_{FLE}$ and $P_{FLE}$ being slightly lower in the SAMS group. Post-hoc analyses revealed statistically significant within-group improvements only in the SAMS group following statin withdrawal, and this was true

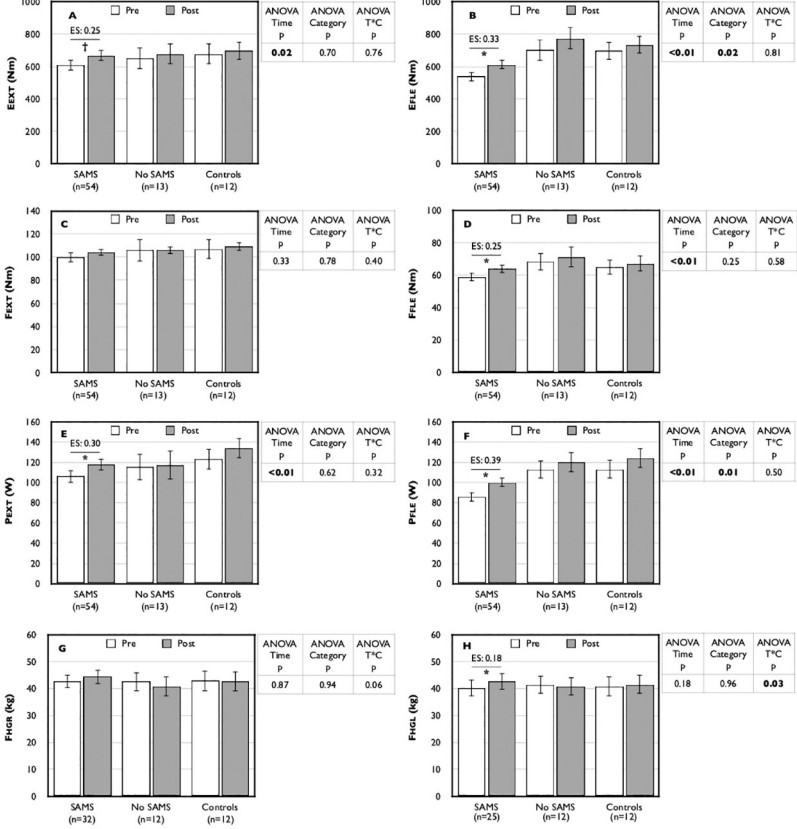

**Fig 2. Pre- and post-statin withdrawal measures of physical performance.** Data are expressed as mean ± standard error; Post value with * is statistically different from pre value with p≤0.01; Post value with † is statistically different from pre value with p<0.05; $E_{EXT}$: endurance in extension (panel A); $E_{FLE}$: endurance in flexion (panel B); $F_{EXT}$: force in extension (panel C); $F_{FLE}$: force in flexion (panel D); $F_{HGL}$: hand grip force left (panel H); $F_{HGR}$: hand grip force right (panel G); $P_{EXT}$: power in extension (panel E); $P_{FLE}$: power in flexion (panel F); SAMS: statin-associated muscle symptoms.

**Table 2. Proportion of patients who display clinically relevant performance improvements following statin withdrawal.**

|  | SAMS n / total (%) | No SAMS n / total (%) | Category p |
|---|---|---|---|
| E<sub>EXT</sub> | 23/54 (42.6) | 2/13 (15.4) | **0.05** |
| E<sub>FLE</sub> | 28/54 (51.9) | 4/13 (30.8) | 0.17 |
| F<sub>EXT</sub> | 9/54 (16.7) | 1/13 (7.69) | 0.38 |
| F<sub>FLE</sub> | 13/54 (24.1) | 2/13 (15.4) | 0.49 |
| P<sub>EXT</sub> | 22/54 (40.7) | 1/13 (7.69) | **0.01** |
| P<sub>FLE</sub> | 29/54 (53.7) | 3/13 (23.1) | **0.04** |
| F<sub>HGR</sub> | 6/32 (18.8) | 0/12 (0.00) | **0.04** |
| F<sub>HGL</sub> | 7/25 (28.0) | 0/12 (0.00) | **0.01** |

Data are expressed as the number of patients with a 15% improvement in performance following statin withdrawal / total number of patients in the considered group; p-values are shown here by a chi-squared test; E$_{EXT}$: endurance in extension; E$_{FLE}$: endurance in flexion; F$_{EXT}$: force in extension; F$_{FLE}$: force in flexion; F$_{HGL}$: hand grip force left; F$_{HGR}$: hand grip force right; P$_{EXT}$: power in extension; P$_{FLE}$: power in flexion; SAMS: statin-associated muscle symptoms.

for five of the six isokinetic dynamometer measures (improvement range +8.8 to 16.6%, [ES: 0.25 to 0.39, all small effects]).

In terms of handgrip performance, a significant interaction was observed for left force (F$_{HGL}$) and this reached near-statistical significance (p = 0.06) for right force (F$_{HGR}$). In both cases, the greatest increase in force following statin withdrawal was seen in the SAMS group (+4.0% for F$_{HGR}$ [ES: 0.09, trivial effect]; +6.2% for F$_{HGL}$, [ES: 0.18, trivial effect]), while values decreased in the No SAMS group (-4.2% for F$_{HGR}$ [ES: -0.16, trivial effect]; -1.7% for F$_{HGL}$ [ES: -0.06, trivial effect]). Indeed, if repeated-measure analyses are done without the Control group, the time x category interactions become statistically significant both for the right and left hands (all p = 0.02).

In order to establish whether observed changes in muscle functions are of clinical relevance, we codified as clinically significant any improvement greater or equal to 15% from baseline for any measure in a given individual [31]. The number of patients from the two statin-using groups showing such improvements is reported in Table 2, and contingency analyses revealed that for five of the eight physical performance measures, the number of patients who experienced a clinical improvement in performance following statin weaning was statistically higher in the SAMS group.

### 3.5. Targeting "real" sufferers of SAMS

For a subset (n = 22) of SAMS-reporting patients, available data allowed us to assess the likelihood of their symptoms being truly caused by statins using the SAMS-CI [16]. Given very low numbers in the "unlikely" category, they were pooled with the "possible" category for analyses. Subjective perception of symptoms resolution was greater in the patients classified as "probable" for suffering from true SAMS using this classification scheme (Fig 3). However, despite improvement in five of eight measures of physical performance following statin withdrawal, we did not observe any significant difference between categories or category x time interactions in repeated measures analyses (Fig 4).

### 4. Discussion

In the face of frequent reports of SAMS [4–7], few objective data have shown negative impacts of statins on muscle performance. Here we present evidence for modest but relevant

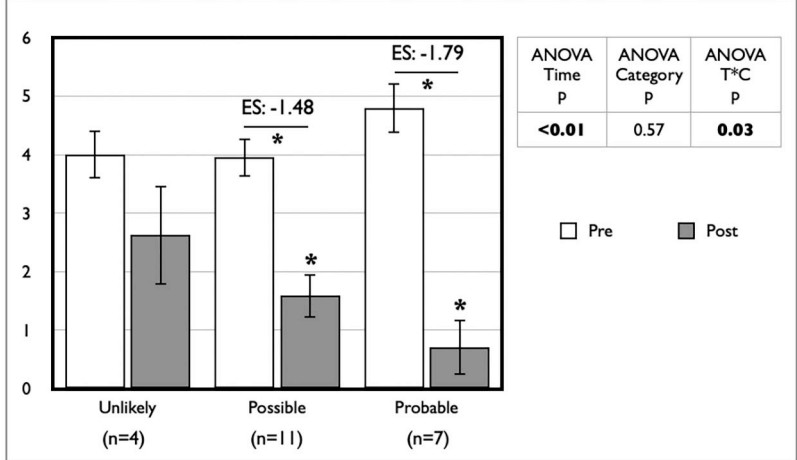

**Fig 3. Impact of statin withdrawal on perceived SAMS intensity by SAMS-CI category using a visual analog scale from 0 to 10.** Data are expressed as mean ± standard error; Post value with * is statistically different from pre value; SAMS: statin-associated muscle symptoms; SAMS-CI: SAMS–clinical index.

improvements on muscle function following statin withdrawal in patients self-reporting SAMS. Indeed, despite the lack of change in biochemical markers or clinical indication of tissue damage, we observe for the entire cohort an overall improvement over time following withdrawal in muscle functions for several objective measures of knee extension and flexion (Fig 2). However, upon closer inspection, the greatest improvements were observed in the SAMS group; in fact, within-group analyses showed no statistically significant changes in the No SAMS and Control groups over time. Although the impact of SAMS on performance remains debatable in the literature, our data aligned with those of Parker et al. (2013) that showed differences in leg strength during isokinetic movements in extension at 60˚/s and 180˚/s and flexion at 60˚/s between atorvastatin-treated participants with or without muscle complaints [22]. It is perhaps not surprising that we observed an impairment in knee extension and flexion performances, as it has been shown that SAMS tend to affect large muscle groups such as the quadriceps [17]. Nevertheless, and unlike that was reported by Parker et al. (2013) [22], we also observed significant though modest improvements in handgrip strength in the SAMS reporting group.

Using a cut-off value (15%) which can be considered clinically relevant and a value greater than what is expected from test habituation, we showed a clearly greater proportion of patients showing improvement in objective measures of muscle performance when reporting SAMS (Table 2). Though the average values of the improvements are modest, they are potentially important because they result in a decrease in the potential to perform daily activities. Also, as muscle performance decreases with aging, the impact of such a decrease in performance with statin use could be clinically more significant in this population, which certainly merits further examination.

Our data must, of course, be nuanced in the context of a growing body of evidence for a large nocebo effect in patients self-reporting SAMS [12–14, 32]. For example, the work of Howard et al. (2021) showed that most symptoms induced by statins were nocebo [32]. In the study, the investigators set up a 12-month multi-crossover trial in which 46 patients in primary CV prevention and 14 patients in secondary CV prevention (65.5±8.6 yrs.) were randomized in three conditions: statin, placebo, and no treatment. Every day, patients were asked to rate

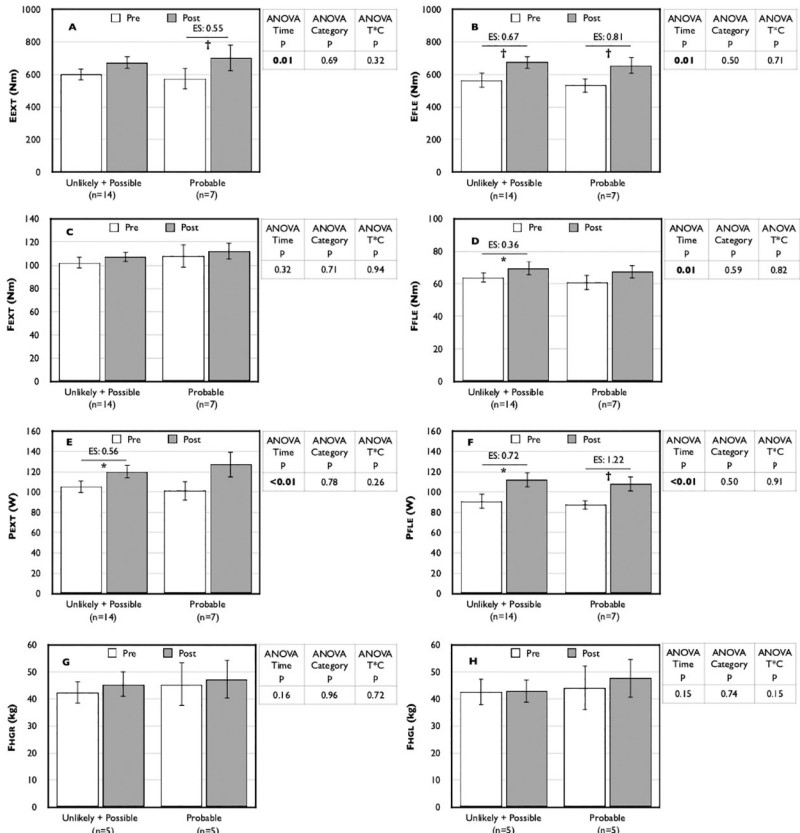

**Fig 4. Impact of statin withdrawal on physical performance by SAMS-CI category.** Data are expressed as mean ± standard error; Post value with * is statistically different from pre value with p≤0.01; Post value with † is statistically different from pre value with p<0.05; EEXT: endurance in extension (panel A); EFLE: endurance in flexion (panel B); FEXT: force in extension (panel C); FFLE: force in flexion (panel D); FHGL: hand grip force left (panel H); FHGR: hand grip force right (panel G); PEXT: power in extension (panel E); PFLE: power in flexion (panel F); SAMS: statin-associated muscle symptoms; SAMS-CI: SAMS–clinical index.

the intensity of statin-associated symptoms (SAS). Results notably showed differences in self-reported SAS intensity between the statin vs. no treatment and placebo vs. no treatment conditions (p<0.01), though no difference between the placebo and statin conditions was observed (p = 0.39). Though their work studied patients who reported SAS, it did not specifically focus on SAMS. Whether or not our patients suffered of "true" SAMS or a confounding nocebo effect, our data showing an objective impact of statin withdrawal on self-reported symptoms intensity and objective measures is of important potential clinical relevance. Indeed, recognizing that a patient could be suffering, objective impacts on muscle functions could and should be considered in treatment assessment and follow up.

It is of great importance to develop or validate tools that purport to identify likely sufferers of "true" SAMS [15]. In this regard, a subset of our data allowed us to classify participants according to the SAMS-CI, a tool that has been proposed as potentially useful in a clinical setting but has yet to be validated [16]. Using this approach, while we did observe a greater improvement in subjective symptoms resolution in those classified as "probably" suffering from SAMS (Fig 3), we did not see any difference in objective measures between categories (Fig 4). These results should be interpreted with great caution as our analyses are weakened by a relatively low number of participants. Nevertheless, the demonstration of a greater subjective

impact in the "probably" group does warrant further study on the potential usefulness of the SAMS-CI.

This study has several strengths. First, the lipid data indicate that we can be confident about compliance with statin withdrawal. Second, the inclusion of a No SAMS group and a Control group allows a more rigorous analysis and interpretation of results, even if these groups are somewhat smaller. Third, the use of validated and standardized sensitive objective measures allows us to discern little but real effects in muscle function. This study is also, to our knowledge, the first work to attempt to validate the SAMS-CI. Finally, the population recruited here represents a typical population of primary CV prevention patients treated with statins, which reinforces the clinical relevance of our data.

A limitation of this study is that recruitment was based on self-reporting of SAMS. Even if we limited the sources of bias using various recruitment criteria, we cannot be certain of the potential influence of the nocebo effect on our results. On the other hand, considering the growing interest in patient-reported outcomes in clinical studies and especially their unique ability to be a true reflection of a "patient-centered approach" [33], further insights into how individuals experience their SAMS and respond to quantitative changes in objective clinical measures can enrich and clarify the evidence from quantitative measures. This knowledge could be used in the future to develop intervention strategies and lifestyle advice tailored to the individual needs of people receiving statin therapy. Another potential cofounder is that the SAMS group was older and presented greater markers of adiposity, which could have influenced some of the results. While the sample studied in this report is typical of the patients' population at our lipid clinic, and thus includes a variety of statins and doses, we cannot exclude differential impacts of these varying formulations on our measures. Also, although all patients were sedentary and were instructed not to change their physical activity habits during the protocol, this was not objectively monitored, which may be a source of intra- and inter-patient variability. In addition, the relatively small number of participants, particularly in the control and No SAMS groups, and especially in the SAMS-CI sub-analysis, limits our statistical power and could have masked the significance of some of the results. The small number of participants also did not allow for rigorous analyses by sex (although in the SAMS group, baseline analyses did not indicate sex differences in response to drug withdrawal; data not shown) or allow us to assess the potential impact of confounding factors using approaches such as multivariable analyses. Furthermore, all participants were Caucasian, potentially limiting application to other ethnicities. While we also limited the recruitment to patients between 30 and 60 yrs of age, primarily to avoid confounding effects of conditions such as sarcopenia on muscle performance measures, this reduces the generalizability of our findings. Our protocol is limited to a short follow-up of two months with two visits (pre- and post-drug withdrawal). A longer follow-up could have led to different conclusions. Finally, our protocol does not allow us to explore or discuss the mechanisms of "true" SAMS.

## 5. Conclusions

Whether or not our study population suffered from "true" SAMS or nocebo effects, our data indicate that participants who self-reported SAMS had improved physical function concurrent with decreased subjective symptom intensity following drug withdrawal. Although the negative impacts of statins on muscle function appear small, these could nonetheless have real and substantive effects in certain patients or patient populations. For example, in patients who have experienced sarcopenia or dynapenia, such as the frail elderly or others suffering from various muscle diseases, an additional loss of even a small portion of their functional physical capacity could contribute to important loss of independence and health-related quality of life

(HRQoL). We suggest that our results should be considered by clinicians in the evaluation and follow-up of treatment with statins in such populations. Even for healthier populations, such as the patients in primary prevention of the present study (who maintained low-to-moderate Framingham risk scores even after statin withdrawal), clinical teams should consider the potential benefits of ending statin therapy or switching to another treatment option in those reporting significant SAMS as this could impact their physical function. Overall, we suggest that careful follow-up of SAMS and their impacts on patient perceived and objective muscle function are warranted so that the clinicians can adjust their advice and prescription to avoid loss of function while maintaining compliance with treatment to effectively prevent the occurrence of CVD.

Clearly, the present results need to be validated or recreated in other studies. More prospective data are needed, perhaps in the context of a randomized double-blinded study. Given the relative ease with which handgrip strength can be measured, the present data certainly warrant future studies on its potential clinical value in assessing functional changes with statin use in various populations. Furthermore, these results need to be examined in other statin-using populations, for example older patients or those in secondary prevention. It also remains of great interest to further efforts to better identify "true" SAMS sufferers and to assess whether functional impacts are greater in this group. Finally, while we focus in this report on objective measures of muscle functions, other impacts of self-reported SAMS such as those on HRQoL need further study.

## Supporting information

**S1 Table. Pre- and post-statin withdrawal plasma variables.** SAMS: statin-associated muscle symptoms; FFA: free fatty acids; FBGL: fasting blood glucose level; INS: insulin; TSH: thyroid-stimulating hormone; PTH: parathormone; eGFR: estimated glomerular filtration rate; CRP: C-reactive protein; RF: rheumatoid factor; Cr: creatinine; UR: urea; PA: pyruvic acid; LAC: lactic acid; LDH: lactate dehydrogenase; ALKP: alkaline phosphatase; Ca: calcium; Na: sodium; K: potassium; Mg: magnesium; NH4: ammonia; P: phosphorus; Cl: chloride; Data are expressed as mean ± standard deviation; Post value with * is statistically different from the pre value with $p \leq 0.01$; Post value with † is statistically different from the pre value with $p < 0.05$. (DOCX)

**S2 Table. Clinical profile of statin users.** SAMS: statin-associated muscle symptoms; * Framingham Score: values are means ± standard deviation and ANOVA $p = 0.06$. (DOCX)

**S1 Checklist. TREND statement checklist.** (PDF)

**S1 Data.** (XLSX)

**S1 Protocol.** (PDF)

## Author Contributions

**Conceptualization:** Jérôme Frenette, Jean Bergeron, Denis R. Joanisse.

**Data curation:** Jérôme Frenette, Jean Bergeron, Denis R. Joanisse.

**Formal analysis:** Paul Peyrel, Jérôme Frenette, Claire Huth, Jean Bergeron, Denis R. Joanisse.

**Funding acquisition:** Jérôme Frenette, Jean Bergeron, Denis R. Joanisse.

**Investigation:** Jérôme Frenette, Nathalie Laflamme, Karine Greffard, Sébastien S. Dufresne, Claire Huth, Jean Bergeron, Denis R. Joanisse.

**Methodology:** Jérôme Frenette, Jean Bergeron, Denis R. Joanisse.

**Project administration:** Jérôme Frenette, Nathalie Laflamme, Karine Greffard, Sébastien S. Dufresne, Jean Bergeron, Denis R. Joanisse.

**Resources:** Pascale Mauriège, Jérôme Frenette, Jean Bergeron, Denis R. Joanisse.

**Supervision:** Nathalie Laflamme, Karine Greffard, Sébastien S. Dufresne, Jean Bergeron, Denis R. Joanisse.

**Validation:** Jérôme Frenette, Claire Huth, Jean Bergeron, Denis R. Joanisse.

**Visualization:** Jérôme Frenette, Jean Bergeron, Denis R. Joanisse.

**Writing – original draft:** Paul Peyrel, Pascale Mauriège, Jérôme Frenette, Jean Bergeron, Denis R. Joanisse.

**Writing – review & editing:** Paul Peyrel, Pascale Mauriège, Jérôme Frenette, Nathalie Laflamme, Karine Greffard, Sébastien S. Dufresne, Claire Huth, Jean Bergeron, Denis R. Joanisse.

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
