## [Decision Letter · Decision Letter 0]

27 Mar 2023

PONE-D-23-01399Impact of statin withdrawal on perceived and objective muscle functionPLOS ONE

Dear Dr. Joanisse,

Thank you for submitting your manuscript to PLOS ONE. After careful consideration, we feel that it has merit but does not fully meet PLOS ONE’s publication criteria as it currently stands. Therefore, we invite you to submit a revised version of the manuscript that addresses the points raised during the review process.

We look forward to receiving your revised manuscript.

Kind regards,

Yoshihiro Fukumoto

Academic Editor

PLOS ONE

Reviewers' comments:

Reviewer's Responses to Questions

**Comments to the Author**

1. Is the manuscript technically sound, and do the data support the conclusions?

Reviewer #1: Yes

Reviewer #2: Yes

2. Has the statistical analysis been performed appropriately and rigorously? 

Reviewer #1: Yes

Reviewer #2: Yes

3. Have the authors made all data underlying the findings in their manuscript fully available?

Reviewer #1: Yes

Reviewer #2: Yes

4. Is the manuscript presented in an intelligible fashion and written in standard English?

Reviewer #1: Yes

Reviewer #2: Yes

5. Review Comments to the Author

Reviewer #1: A clinical trial was conducted which aimed to assess if subjective and objective measures of muscle function improve after statin withdrawal in patients reporting statin-associated muscle symptoms (SAMS). Both subjective and objective measures of muscle function showed moderate improvement.

Minor revisions:

1- Indicate the date range subjects were enrolled in the study.

2- Define the abbreviation SD.

3- Lines 203-207: Provide complete sample size calculation details such that the numbers can be verified. The power calculation should include: (1) the estimated outcomes in each group; (2) the α (type I) error level; (3) the statistical power (or the β (type II) error level); (4) the target sample size and (5) for continuous outcomes, the standard deviation of the measurements.

4- Line 213: State the underlying covariance structure used in the mixed models.

5- The wording in line 216 is awkward. Consider using the following statement instead. “P-values less than 0.05 were considered statistically significant.”

6- The standard statistical term for average is mean.

7- If the interaction effect is significant, provide an interpretation of the results, but do not test main effects because the tests for main effects are uninteresting in light of significant interactions. If interaction effects are non-significant, drop the interaction effects from the model and test the main effects. Determining which results to present when testing interactions is often a multi-step process.

Reviewer #2: This is an interesting study to evaluate the relationship between statin withdrawal and perceived and objective muscle function. I have a few comments to the authors.

1. I think many abbreviations such as ENEXT, ENFLE, FOEXT, FOFLE, POEXT, POFLE, FOHGR, FOHGL etc. are not familiar to the readers. At least, in Tables and Figures the authors need to explain (spell out?) each parameter.

2. What is the clinical implication? Some patients with SAMS will recover their muscle function after long-term statin treatment.

3. Fig.2-4: No adjustment for baseline characteristics in each parameter. Could the authors provide comparison data with multivariable adjustment?

6. PLOS authors have the option to publish the peer review history of their article (what does this mean?). If published, this will include your full peer review and any attached files.

Reviewer #1: No

Reviewer #2: No

---

## [Author Response · Author response to Decision Letter 0]

26 Apr 2023

Response to editor

We would like to thank you for your helpful and encouraging comments. We hope that our responses below and the changes to the manuscript will allow acceptance of the revised work.

Response to reviewers

We would like to thank the reviewers for their helpful and insightful comments. We hope that they will find that our responses below and changes to the manuscript warrant acceptance of the revised work. 

Reviewer #1:

A clinical trial was conducted which aimed to assess if subjective and objective measures of muscle function improve after statin withdrawal in patients reporting statin-associated muscle symptoms (SAMS). Both subjective and objective measures of muscle function showed moderate improvement.

1- Indicate the date range subjects were enrolled in the study.

Response: This has been added to the methods on page 6.

2- Define the abbreviation SD.

Response: This has been done (page 5 and S1 and S2 Tables).

3- Lines 203-207: Provide complete sample size calculation details such that the numbers can be verified. The power calculation should include: (1) the estimated outcomes in each group; (2) the α (type I) error level; (3) the statistical power (or the β (type II) error level); (4) the target sample size and (5) for continuous outcomes, the standard deviation of the measurements.

Response: We have clarified this as requested in the methods (page 10).

4- Line 213: State the underlying covariance structure used in the mixed models.

Response: This has been added to the methods (page 10).

5- The wording in line 216 is awkward. Consider using the following statement instead. “P-values less than 0.05 were considered statistically significant.”

Response: This has been adjusted in the methods section (page 10).

6- The standard statistical term for average is mean.

Response: This has been corrected throughout the manuscript.

7- If the interaction effect is significant, provide an interpretation of the results, but do not test main effects because the tests for main effects are uninteresting in light of significant interactions. If interaction effects are non-significant, drop the interaction effects from the model and test the main effects. Determining which results to present when testing interactions is often a multi-step process.

Response: We have adjusted our presentation of the results as requested (pages 11 to 19).

Reviewer #2:

This is an interesting study to evaluate the relationship between statin withdrawal and perceived and objective muscle function. I have a few comments to the authors.

1. I think many abbreviations such as ENEXT, ENFLE, FOEXT, FOFLE, POEXT, POFLE, FOHGR, FOHGL etc. are not familiar to the readers. At least, in Tables and Figures the authors need to explain (spell out?) each parameter.

Response: We have changed the abbreviations of the performance measures to better reflect common usage and readability. Changes have been made throughout the manuscript and tables and figures.

2. What is the clinical implication? Some patients with SAMS will recover their muscle function after long-term statin treatment.

Response: This has been expended upon in the conclusion (page 24).

3. Fig.2-4: No adjustment for baseline characteristics in each parameter. Could the authors provide comparison data with multivariable adjustment?

Response: We are aware that this is a limitation of the presented work and have added a sentence in the limitations section to this effect (page 23). It certainly would be of interest to assess the impact of factors such as age, sex, or activity levels on the outcomes that were measured, however, we fear that the sample size we have cannot support such analyses without major loss of statistical power. The sample size was in fact calculated only to meet the primary objective of the manuscript, i.e., the effects of statin withdrawal on perceived SAMS and objective muscle performance, without adjustments.

Nevertheless, in our statistical model subject ID is included as a within-subjects factor, allowing adjustment for initial value of measured parameters.

---

## [Decision Letter · Decision Letter 1]

9 May 2023

Impact of statin withdrawal on perceived and objective muscle function

PONE-D-23-01399R1

Dear Dr. Joanisse,

We’re pleased to inform you that your manuscript has been judged scientifically suitable for publication and will be formally accepted for publication once it meets all outstanding technical requirements.

Kind regards,

Yoshihiro Fukumoto

Academic Editor

PLOS ONE

Additional Editor Comments (optional):

Reviewers' comments:

Reviewer's Responses to Questions

**Comments to the Author**

1. If the authors have adequately addressed your comments raised in a previous round of review and you feel that this manuscript is now acceptable for publication, you may indicate that here to bypass the “Comments to the Author” section, enter your conflict of interest statement in the “Confidential to Editor” section, and submit your "Accept" recommendation.

Reviewer #1: All comments have been addressed

Reviewer #2: All comments have been addressed

2. Is the manuscript technically sound, and do the data support the conclusions?

Reviewer #1: (No Response)

Reviewer #2: Yes

3. Has the statistical analysis been performed appropriately and rigorously? 

Reviewer #1: (No Response)

Reviewer #2: Yes

4. Have the authors made all data underlying the findings in their manuscript fully available?

Reviewer #1: (No Response)

Reviewer #2: (No Response)

5. Is the manuscript presented in an intelligible fashion and written in standard English?

Reviewer #1: (No Response)

Reviewer #2: (No Response)

6. Review Comments to the Author

Reviewer #1: (No Response)

Reviewer #2: (No Response)

7. PLOS authors have the option to publish the peer review history of their article (what does this mean?). If published, this will include your full peer review and any attached files.

Reviewer #1: No

Reviewer #2: No

---

## [Editor Report · Acceptance letter]

6 Jun 2023

PONE-D-23-01399R1 

Impact of statin withdrawal on perceived and objective muscle function 

Dear Dr. Joanisse:

I'm pleased to inform you that your manuscript has been deemed suitable for publication in PLOS ONE. Congratulations! Your manuscript is now with our production department. 

Kind regards, 

on behalf of

Dr. Yoshihiro Fukumoto 

Academic Editor

PLOS ONE